

# Measurement report:
# Atmospheric mercury measurements at the Russian Arctic station Amderma and connection with eruptions of Icelandic volcanoes

Fidel Pankratov[a,*], Alexander Mahura[b], Oleg Katz[c], Tuukka Petäjä[b], Valentin Popov[d], Vladimir Masloboev[a]

[a]Institute of Northern Environmental Problems, Kola Science Center, Russian Academy of Sciences, Fersman Str. 14A, Apatity, 184200, Russia.
[b]Institute for Atmospheric and Earth System Research (INAR)/Physics, Faculty of Science, University of Helsinki (UHEL),
P.O. Box 64, Helsinki, FI-00560, Finland.
[c]Continuing Education of the Bar (CEB), University of California, Oakland, California.
[d]Research and Production Association "Typhoon" of Roshydromet, Pobedy Str. 4, Obninsk, 249038, Russia.

*Correspondence to: Fidel Pankratov (fidelpankratov@gmail.com)*

**Abstract.** Mercury (Hg) is a toxic substance and accumulates in the biosphere causing negative impacts to the well- being
of flora and fauna as well as humans. In this study, we analysed the long-term time-series (2001- 2013) of the gaseous elemental mercury measurements at the Arctic station Amderma (Russia). We explored the influence of long-range atmospheric transport of gas-phase mercury into the Arctic from the volcanic eruptions in Iceland in 2010-2011. The change in the dynamics of atmospheric Hg concentration was identified. Contrasting time periods of 2001–2009 and 2010–2012 periods, we quantified a negative trend of -0.66 ng for the earlier period and a positive trend of +0.97 ng for the latter
period. Our analysis highlighted that the elevated Hg concentrations at Amderma were associated with active volcanic eruptions in Iceland, namely Eyjafjallajökull and Grímsvötn in 2010 and in 2011, respectively. The observed Hg concentrations were in the range of $1.81 \div 2.58$ ng m$^{-3}$ in Apr-Jun 2010 and $1.81 \div 3.31$ ng m$^{-3}$ in May-Jun 2011 compared with the annual average Hg concentrations of $1.51 \pm 0.41$ ng m$^{-3}$. This is the first time to detect such an elevated Hg concentration during the active volcanic eruptions measured over 3200 km away from the eruption source. The calculated
atmospheric backward trajectories (at altitudes of 500, 1500 and 3000 meters above sea level) underlined the occurrence of the Hg elevated concentrations and confirmed the atmospheric transport from the areas of these two volcanoes. Therefore, it can be assumed that these volcanoes were the main source of the increased Hg concentrations at the Amderma station resulted due to the long-range atmospheric transport of the volcanic emissions.

**Keywords:** Amderma station; Russian Arctic; Long-term atmospheric mercury measurements; Volcanic eruptions in Iceland; HYSPLIT trajectory modeling; gaseous mercury.

## 1  Introduction

Mercury is bioaccumulative and toxic to the environment (Sonke et al. 2013). The mercury in the Arctic is due to global mercury emissions and their transport to high latitudes (Dastoor et al. 2022a; Dastoor et al. 2022b). The transported
mercury is deposited to terrestrial environments and into the Arctic Ocean. The evasion of Hg$^0$ from the oceans is balanced by the total oceanic deposition of Hg$^{II}$ from the atmosphere. The mechanisms whereby reactive Hg species are reduced to volatile Hg$^0$ in the oceans are poorly known, but reduction appears to be chiefly biological (Mason et al. 1994). Lateral transport through rivers (Sonke et al. 2018) is important, as well as cryosphere emissions (Araujo et al. 2022; Yue et al. 2023). Volcanic eruptions are also an important source for atmospheric mercury (Ariya et al. 2015). It is also necessary to
have accurate data on the amounts of different forms of Hg from the volcanic emissions so that these values can be used as data for mathematical models describing the lifecycle of Hg in the Earth System (Lee et al. 2001; Ryaboshapko et al. 2002).



Volcanic emissions disperse on long distances. For example, the Eyjafjallajokull volcano (Iceland) on 13 Apr and 17-21 Apr 2010 (Flentje at al. 2010) resulted in elevated concentrations of the particulate fractions (with particle size more than 3

nm) were recorded at the global atmospheric monitoring stations (Zugspitze/ Hohenpeissenberg - Germany, 2650 m above sea level, asl). Volcanoes are the main natural sources of Hg entered into the atmosphere (Krabbenhoft and Schuster, 2002). The volcanogenic Hg flux from passively degassing volcanoes is about 30 t yr$^{-1}$. The flux from erupting volcanoes is much larger, and it is about 800 t yr$^{-1}$, and geothermal sources contribute to the atmosphere roughly 60 t yr$^{-1}$ (Varekamp and Buseck, 1986) however, other authors considered that the emission is not so significant and amounts to not any more than

1.3 t yr$^{-1}$ (Pirrone et al. 2001). Therefore, current estimates of the global Hg emissions suggest that the overall contribution from natural and anthropogenic sources is nearly 2320 t yr$^{-1}$ (Pirrone et al. 2010). Measurements of Hg speciation at the crater edge of the volcanoes Masaya (Nicaragua) showed that over 90% of Hg was presented in its elemental form. The particulate and reactive gaseous fractions comprised 1-5% and ~1%, respectively, of the total Hg (Witt  et al. 2008). The contributions from primary natural sources of Hg (estimated as ~500 t yr$^{-1}$) remain as a subject of considerable uncertainty.

This is particularly the case for volcanic activity, which is considered an important but poorly understood source of Hg to the environment (Edwards et al. 2021). Integrated over the past 270 years of ice core history, anthropogenic inputs accounted -52%, volcanic events - 6%, and background sources - 42%. More importantly, over the past 100 years, anthropogenic sources contributed about 70% of the total mercury input. (Krabbenhoft and Schuster 2002). Overall, considerable uncertainties remain in the lifecycle of Arctic atmospheric mercury and particularly monitoring in sensitive

environments is needed (Dastoor et al. 2022b).

Many studies of global volcanogenic mercury (Hg) emissions showed that, in general, the transport of atmospheric mercury from the volcanic activity can be estimated relatively to other compounds that are formed during  degassing and eruption stages of volcanoes. During a volcanic eruption about 45 different trace elements from the Earth's crust can be emitted into the atmosphere. However, there is no yet direct correlation between the amounts of the trace elements emitted into the

atmosphere during the eruption and the number of reported cases of the volcanic eruptions (Mambo and Youhida, 1993).The volcanic eruptions and their carbon emissions to have driven severe environmental events in the geological past. Applying excess mercury loading to estimate large igneous province-associated carbon emissions, revealing that multi-millennial episodes of activity plausibly drove recognized and temperature increases that demonstrates mercury's potential as a tool to resolve past carbon fluxes (Fendley et al. 2024).

Some studies show that continuously degassing volcanoes emit up to several hundred tons of SO$_2$ per day during dormancy, while total gas emissions (~H$_2$O+ CO$_2$+ SO$_2$) are at least an order of magnitude greater and can reach values is about 27÷55 t yr$^{-1}$ (Etna, Masaya or Satsuma-Iwo Jima). Accordingly, the question arises whether passive degassing can trigger some of the processes that can lead to an eruption (Girona T. et al. 2015). Mercury levels in volcanic fumaroles in combination with sulfur analyses and sulfur dioxide (SO$_2$) flux data, allowed to estimate the global Hg flux. The average contribution of SO$_2$

to the atmosphere by volcanoes, estimated largely by extrapolation from direct measurements of volcanic SO$_2$, is about 18 t yr$^{-1}$. The estimated 6 t yr$^{-1}$ (36%) is from non-erupting degassing volcanoes  (Stoiber et al. 1987). SO$_2$ contributes 64% to the global volcanic sulfur flux of 10 t yr$^{-1}$. (Andres and Kasgnoc, 1998). However, we can use the directly measured SO$_2$ fluxes and known molar ratios (e.g., H$_2$S/SO$_2$) for a semi-quantitative estimate of other gas components emitted (e.g., H$_2$S). The total annual emission of HCl is 1÷170 t yr$^{-1}$ (Halmer et al. 2002). For active volcanoes, the average Hg/S ratio value

resulted is 1.2×10$^{-7}$ and the mercury emissions from the Stromboli volcano (Italy) were estimated in the range of 1.3÷5.5 kg yr$^{-1}$ (Ferrara et al. 2000). After correcting for 'unmeasured' SO$_2$ emissions, the estimated total global flux of Hg into the atmosphere is about 112 t yr$^{-1}$. At the same time, new data on volcanic emissions Hg were obtained on the basis of satellite sounding of sulfur dioxide (SO$_2$) - 232 t yr$^{-1}$. It should also be noted that more than 90% of volcanic mercury emissions occur in various latitudes of the Northern Hemisphere, 1.8 times higher than in the Southern Hemisphere (Geyman B. M. et



al. 2023). There are regional differences in average emissions as shown by Nriagu and Becker (2003) from analysis of a
long-term (20 years) dataset. Previous estimates for Hg, based on limited measurements from the volcanic plumes, span
three orders of the magnitude ~100 t yr$^{-1}$, or from <1% to ~50% of the total natural Hg emissions (Pyle and Mather, 2003).
According to Bagnato et al. (2009) the gas emissions of the La Soufrière volcano (Italy) had the mean total Hg/H$_2$S mass
ratio (~3.2×10$^{-6}$) of the volcanic plume measured close to the source vent, with the H$_2$S plume flux (~0.7 t day$^{-1}$). Hinkley
et al. (1999) mentioned that there are still open questions regarding the volcanic origin of other elements. Both before the
eruption of the volcano and at the end of the active phase of the eruption, large amounts of gases escape through the cracks
in the Earth's crust an active process of degassing. Research conducted from 1991 to 1995 in the area of the Etna volcano
(Italy) showed that during the eruption the atmospheric concentrations of the elements such as Bi, Cu, Cd, Sn and Zn had
increased (Gauthier and Le Cloarec, 1998). The collected samples of magma showed that the concentrations of Hg and Bi
are approximately equal. Considering that the ratio of Bi/SO$_2$ concentrations is in a range from 10$^{-6}$ to 10$^{-4}$ it can be
assumed that the ratio of Hg/SO$_2$ will be also in the same range. Annual contribution of Bi is about 37 t yr$^{-1}$ from the active
phases of eruptions, and it is about 5 t yr$^{-1}$ from the degassing (Gauthier and Le Cloarec, 1998). Main discussions about
emissions during eruptions are focused on amounts of emissions that are produced by each particular volcano (Mather et al.
2004), and at which ratio the mercury in a gas-phase vs. mercury deposited on aerosol particles in the same volume is
observed. A usage of data retrieved from natural deposition of Hg in peat bogs or ice cores samples (Roos-Barraclough et
al. 2002) or measurements of Hg concentration in a  gas-phase during volcanic eruption (Temme et al. 2003) can provide
more accurate information about Hg flux into the atmosphere (tested this assumption applying trajectory modeling
approach). Large excess degassing is associated with various volcanic activities, including explosive and effusive eruptions.
Excess degassing is an important concept for understanding volatile balance, eruption mechanisms, and crustal magma
differentiation, but the mechanism, source, and causes of excess degassing are not yet fully understood. It should be taken
into account that data on volcanic gas composition are still very limited, particularly for volatiles released during eruptions,
and the initial source of excess degassing is still unclear, particularly for siliceous volcanism (Shinohara, 2008).

The aim of our study was to provide an analysis of the long-term gaseous elemental mercury concentrations from the
Amderma station located in the Russian Arctic. The measurement dataset covered period from Jun 2001 to Dec 2013.
Particularly, we explored the seasonal cycle and trends of the Hg concentrations and identified two potential emission
sources such as the Eyjafjallajökull and Grimsvötn volcanos in Iceland that influenced the observed elevated Hg
concentrations in 2010 and 2011.

## 2  Methodology

### 2.1  Site measurement. Long-term measurements of atmospheric mercury

Since Jun 2001 the long-term monitoring of the gaseous elemental mercury (hereafter, mercury, Hg) in the surface
atmospheric layer was carried out near the Amderma settlement (69$^0$72` N; 61$^0$62` E; Meteorological station WMO-230220,
28 m asl, Yugor Peninsula, Russia). It is located on the shore of the Kara Sea and close to the Arctic border between Europe
and Asia (**Fig 1**).

Amderma settlement is located on the northern tip of the Yugor Peninsula on the coast of the Kara Sea. The Pai-Khoi ridge
occupies the center of the peninsula, and it is a continuation of the Polar Urals. The surrounding Amderma area is swampy
tundra. The vegetation period is short and lasts only 2 months. Amderma is in the subarctic belt, characterized by severe
and variable weather, cloudiness, and long winters with frequent snowstorms. The climate is formed in conditions of low
solar radiation in winter, under the influence of the northern seas and intensive western (from the Atlantic Ocean) moist air
masses transport as well as under influence of local physical-geographical features of the territory. The radiation processes
play a major role in climate formation. Polar day and polar night cause extremely uneven distribution of solar radiation



throughout the year.

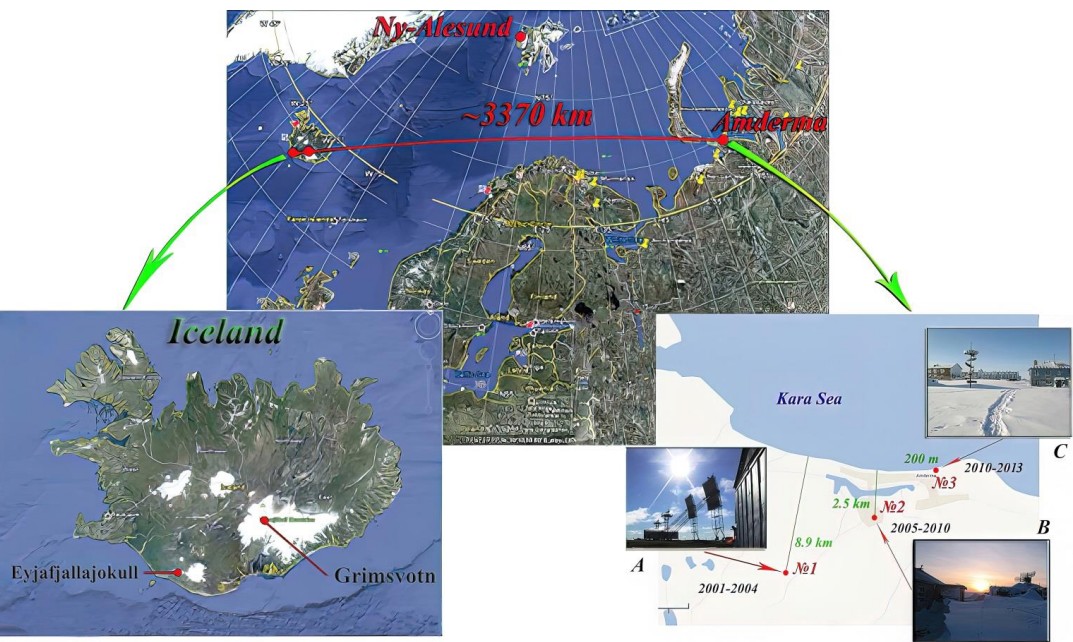

**Figure 1. Geographic layout (© Google Earth 2018; 3 photos (A, B, C) by F. Pankratov) of the Russian Arctic station Amderma (Yugor Peninsula) for the mercury measurements with respect to the sources (i.e., the volcanoes Eyjafjallajökull and Grímsvötn in Iceland).**

The climate is Arctic. Most of the year the sea surface is covered with ice and snow. Winter is slightly softened by the influence of the surrounding sea. It lasts from the end of September until the end of May. The lowest air temperatures are below -40 ℃ and frequently observed during January-March. At the same time, thaws are observed when Atlantic air masses arrived. Summer is short and cool, with average air temperatures ranging from +2.3 to +7.3 ℃. The average annual air temperature is -7 ℃, with the coldest month of February and the warmest month of August. More than 300 mm of precipitation falls annually (with about 30% in November-March), and precipitation is observed for about 200 days. During the year, there are about 290 days with the air humidity over 80%, and there are up to 200 cloudy days. Snow cover remains for most of the year (more than 230 days), and it appears at the end of September (occasionally - by August) and it lasts until the beginning of June (occasionally - by July). There are up to 100 days with snowstorms, which observed during September-June.

During the 13-year (2001-2013) measurement period the analyzer "Tekran 2537A" was located at three points/sites at different distances (i.e., 8.9, 2.9, and 0.2 km) from the coast of the Kara Sea. It should be noted that from September 2015 to July 2016, at the 3rd site the parallel measurements of atmospheric mercury were carried out using the "Tekran 2537A" and "Lumex 915 AM" analyzers. The "Tekran 2537A" instrument hardware complex kept continuous measurements of mercury in the surface layer of the atmosphere. The distance from the ground up to the inlet filter ranged from 5 to 8 m.

The measurement complex included the following equipment: (1) Mercury Vapor Primary Calibration Unit "Tekran 2505"; (2) Tekran Model 1100 Zero Air Generator; (3) Tekran Model 1120 Standard Additional Controller; and (4) Automated Ambient Air Analyzer "Tekran 2537A".

The analyzer exhibits the following characteristics: high temporal resolution, with adjustable regimes ranging from 5 minutes to 200 hours, and an air pumping capacity of 0.7 to 1.5 liters per minute. Its detection limit is 0.11 ng m$^{-3}$, with a measurement error of ±10% (source: Tekran Model 2537 CVAFS Automated Mercury Analyzer). Mercury sorption is facilitated using a triple nine gold trap. The device features automated internal calibration, ensuring full autonomy in



measurement and data processing. Mercury, accumulated on a gold sorbent cartridge ("A" or "B"), undergoes thermal desorption and quantification through a cold vapor atomic fluorescence spectrometer. Mercury concentrations in ambient air were measured at 30-minute intervals across two channels. After verifying the data quality, an average of two consecutive measurements (1-hour averages) was calculated, along with daily average mercury concentrations. Sampling, preparation, and analytical methods for atmospheric mercury monitoring were implemented based on EPA Method 1669. This methodology was adapted for background concentration measurements at the station by modifying the automatic calibration period and setting the sorption period to 30 minutes. The reliability of Hg background concentration measurements hinges on precise sampling and the application of cold vapor atomic fluorescence techniques to register detected signals. Calibration posed a critical challenge due to the lack of stable standards for reduced mercury concentrations (ng m³). Consequently, a mercury vapor generator ("Tekran 2505") housed in a thermocontainer was utilized to introduce precise amounts of mercury gas-air mixtures with known concentrations. An internal Hg vapor source complements this external calibration. The "Tekran 1120" standard addition controller enables programming of Hg injection time from the internal source into sampled ambient air. Additionally, the controller supplies clean air to the system via a connection to the "Tekran 1100" clean air generator. This setup ensures accurate calibration and reliable operation (Pankratov et al., 2015).

This analyzer was selected as the primary instrument used for the Hg continuous measurements at all polar stations including Amderma (Steffen et al. 2005). In 1998 the "depletion" of atmospheric mercury in the air (AMDEs - Atmospheric Mercury Depletion Events) has been recorded on the polar station "Alert" (Canada). This event is an abrupt decrease of the mercury concentration in the atmospheric boundary layer in the spring time (Schroeder et al. 1998). Our monitoring at the Amderma station demonstrated that the Hg background concentrations in the surface layer in the Russian Arctic $(1.51\pm0.41$ ng m$^{-3}$) are similar to the global background of $1.51\div1.71$ ng m$^{-3}$ in the Northern Hemisphere (Steffen et al. 2008). Moreover, the atmospheric mercury depletion events (AMDEs) were observed at Amderma each year. During the Hg depletion, the average Hg concentrations usually decreased below 1 ng m$^{-3}$ and with a significant variability (Pankratov et al. 2013).

Note that this Hg monitoring in the surface layer of the atmosphere at the Russian Arctic station Amderma was carried out for the first time. Table 1 shows the spatial distribution (at selected locations) of the Hg concentration in the Northern Hemisphere for years of 2001-2005.

**Table 1**. **Annual average concentrations (ng m$^{-3}$) of the gaseous elemental mercury in the Northern Hemisphere, (Reports of the Russian Academy of Science (RAS), and the Swedish Environmental Research Institute (SERI/ IVL)) (Pankratov et al., 2015).**

| Location | Coordinates | 2001 | 2002 | 2003 | 2004 | 2005 | Source |
|---|---|---|---|---|---|---|---|
| Andoya, Norway | 69$^0$N, 16$^0$E | | | | 1.64 | | IVL, 2010 |
| Pallas, Finland | 68$^0$N, 24$^0$E | 1.41 | 1.48 | 1.57 | 1.49 | 1.63 | IVL, 2010 |
| Ny-Alessund, Norway | 79$^0$N, 12$^0$E | 1.61 | 1.63 | 1.63 | 1.52 | 1.61 | IVL, 2010 |
| Amderma, Russia | 69$^0$N, 61$^0$E | 1.65 | 1.73 | 1.71 | 1.52 | 1.54 | Pankratov et al. 2013 |
| Northern Sea Shipping Route | 64$^0$N, 40$^0$E | 0.32 | | | | | RAS., vol. 322, 2002 |
| Kara Sea, Russia | | | 0.89 | | | | RAS., vol. 322, 2002 |
| Barents Sea, Russia | | | 0.61 | | | | RAS., vol. 322, 2002 |

## 2.2 HYSPLIT trajectory modelling

Backward trajectory modeling is a widely used tool to assess the potential of the atmospheric transport from different geographical locations where natural or anthropogenic sources of Hg may be presented. Determination of spatial positions of the air parcels in the atmosphere during such movements may allow an identification of the potential paths and regions where pollution can be found and how it is related to the potential sources (Mahura at al. 2013). The accuracy of the





trajectory calculation is generally of the order of 20% of the travel distance, although in some cases the ratio may be even higher (Stohl et al. 1998). In our study, the National Oceanic and Atmospheric Administration (NOAA) on-line transport and dispersion Hybrid Single-Particle Lagrangian Integrated Trajectory model (HYSPLIT) v 4.5 model available in an interactive mode was employed (http://www.arl.noaa.gov/ready/open/hysplit4.html; Draxler and Rolph, 2003; Rolph 2003). For the trajectory modeling, the meteorological gridded dataset (http://dss.ucar.edu/pub/reanalysis; NCEP / NCAR global Present – 1948; Kalnay at al. 1996) was utilized as input. Each trajectory was calculated for cases with the Hg elevated

concentrations at the Amderma station. For simplicity, the trajectories arriving at the station at 3 levels (i.e., 500, 1500 and 3000 m asl) were calculated backward in time up to 120 hours (i.e., 5 days). To calculate the trajectories the modeled vertical motion method was used. Note, there is also a difference of 4 hour between the Amderma local standard time (LST) and Coordinated Universal Time (UTC) provided in the dataset and used in the model, respectively. Hence, all trajectories were calculated with a corrected time in order to match the corresponding measurements at Amderma. Then, all

calculated trajectories were attributed to sectors according to the pathways of the atmospheric transport.

## 3   Result and discussions

### 3.1  General features of mercury concentrations in Amderma

The Amderma station was chosen to assess the flux of Hg and persistence organic pollutants (POPs) to the ecosystems of the Russian Arctic. The long-term (2001-2013) time-series of the obtained atmospheric Hg concentrations is shown in **Fig.**

**2**. As mentioned in the section 2.1 the analyzer "Tekran 2537A" was subsequently placed at three points/sites at three different distances (8.9, 2.9 and 0.2 km) from the coast of the Kara Sea.

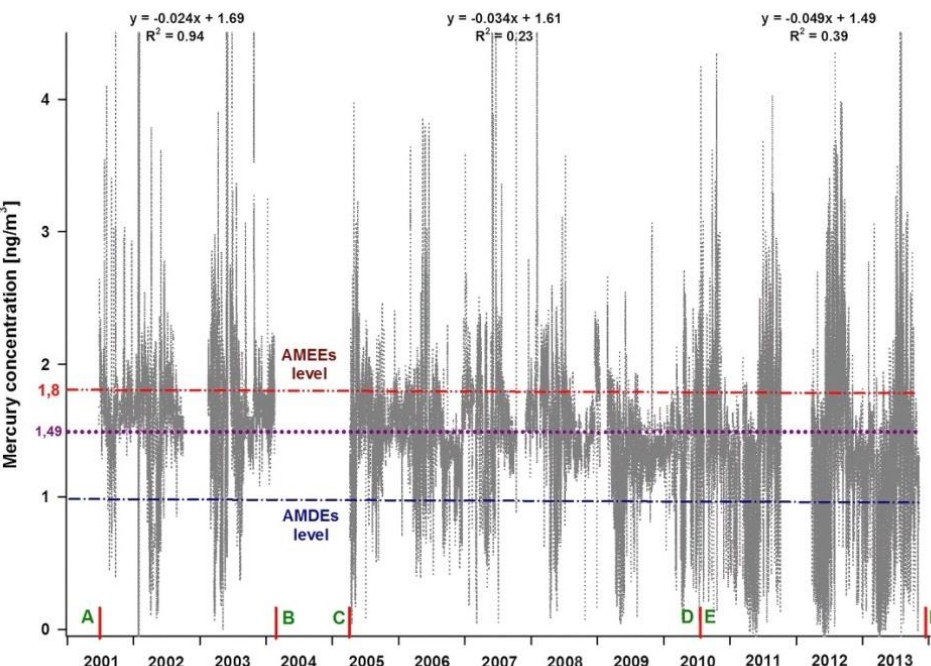

**Figure 2. The long-term time-series of the atmospheric mercury concentration at the Amderma station for the periods: 2001–**
**2004 (A, B), 2005–2010 (C, D), and 2010–2013 (E, F). For the entire period: mercury concentration above the red dash-dotted line with two points is the AMEEs area; mercury concentration below the blue dash-dotted line is the AMDEs area (Pankratov, 2015).**





Analysis of the data measured at site №1 (at distance of 8.9 km from the coast) showed that during 2001–2004 the average

Hg concentration was 1.65±1.91 ng m⁻³ (maximum - 75.51 ng m⁻³ and minimum - 0.11 ng m⁻³, which is the detection limit of the Tekran Instrument (see **Fig. 2, A–B**)). The atmospheric mercury depletion events (AMDEs) occurred at the site №1 were less frequent compared to two other sites (№2 and №3) located closer to the coast (see **Fig. 2, C–D and D–F**). Similarly to AMDEs, the term Atmospheric Mercury Enhancement Events (AMEEs) was used for cases with a relatively long time (at least, 2 hour) elevated concentrations of Hg (>1.81 ng m⁻³). For the site №2 (2.9 km), during 2005–2010, the

average Hg concentration was 1.48±0.42 ng m⁻³ (maximum - 14.53 ng m⁻³). For this site a decreasing trend for the Hg concentration was observed (see **Fig. 2, C–D**). In Jun 2010, the analyzer was installed at the site №3 (0.2 km). Analysis of these data showed that from Jun 2010 to Oct 2013 the average concentration was 1.38±0.84 ng m⁻³ and maximum - 94.35 ng m⁻³ (**Fig. 2, E–F**) as well as seasonal dynamics for the entire observation period (ARCTIC DATASETS as part of PEEX International Collaboration, https://www.atm.helsinki.fi/peex/images/after_2.pdf )

**3.2 Seasonal variability of mercury**

As analysis of the 2001-2013 dataset showed the average Hg concentration was about 1.15±0.41 ng m⁻³. For 2001-2009, there is a decreasing trend in the concentration which is on average is about 1.32±0.31 ng m⁻³ (**Fig. 3**). However, for 2010-2012, there is an increase in the concentration (1.67±0.31 ng m⁻³) by about 26.5%.

The analysis underlined that throughout the entire period of the measurements there were events showing a significant

temporary decrease in the Hg concentration (i.e., below 1.0 ng m⁻³). Such periods, called the AMDEs, are seen in **Fig. 2**. Note these events occurred every year, and mostly took place from the late Mar to the early Jun. Moreover, during AMDEs the smallest variability in the concentration was observed mainly in the winter months.

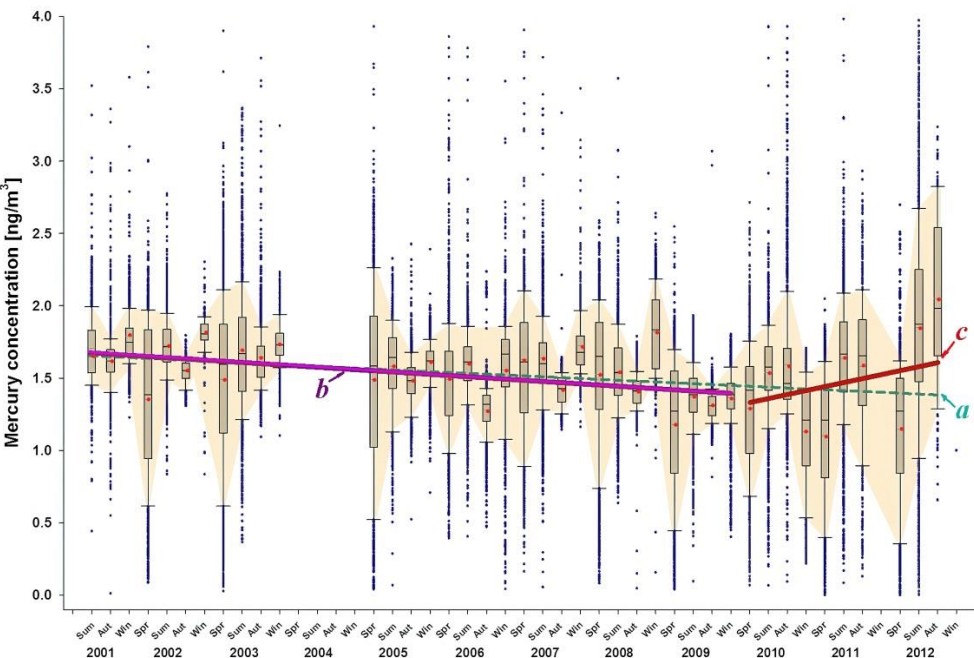

**Figure 3. Seasonal variability of the mercury concentration in the surface layer of the atmosphere at the Amderma station**

**represented by the linear trends for the periods: (a) 2001-2012, (b) 2001-2009, and (c) 2010-2012. Comments: The bars indicate the seasonally averaged data. The red line indicates the median, the boxes depict the quartile ranges and the whispers 5 and 95 percentiles. The blue dots illustrate the statistical outliers. The short black line inside the boxes is the mean concentration for a given season (Pankratov, 2015).**





The dynamics of Hg in the ambient air was not typical in 2010 compared with the previous years (see **Fig. 3**). The elevated Hg concentrations were recorded in spring and summer.  One of the reasons for this behavior could be assigned to the long-range atmospheric transport from Iceland which took place during the eruption of the Eyjafjallajökull volcano. We explored this connection in more details. From 13 Apr to 12 May 2010 the Hg concentrations were significantly higher ($1.81 \div 2.75$ ng m$^{-3}$) than the average long-term values ($1.51 \pm 0.41$ ng m$^{-3}$) and characteristic for the Northern

Hemisphere (Fisher et al. 2012). The analysis of the pathways of the air masses transport (i.e., the calculated backward atmospheric trajectories) from Iceland suggests that in the middle and second half of Apr 2010 the Amderma station was in the area affected by the volcanic emissions, unlike other global stations of the Hg monitoring (e.g., Alert, Canada and Ny-Ålesund, Norway). Note that similar behavior was also observed in 2011 during eruption of the other Icelandic volcano – Grimsvötn. For the 2001-2009 period the negative trend (-0.35 ng per period) was obtained (**see line b in  Fig.**

**3**). However, for 2010-2012 the significant positive dynamics and its high variability were  identified (**see line c in Fig. 3**). On average, the Hg concentrations were $1.43 \pm 0.41$ ng m$^{-3}$ and $1.55 \pm 0.71$ ng m$^{-3}$ in 2010 and 2012, respectively. For this three-year period, the positive trend (+0.12 ng per period) was found. Note that such significant increase in the Hg concentration was registered for the  first time at the Amderma station. Moreover, such behavior is not typical (if the long-term trend in the reduction of the Hg concentration in the atmosphere is considered).

This can be explained by the Icelandic volcanic eruptions of Eyjafjallajökull (in 2010) and of Grímsvötn (in 2011). The volcanic cloud passed over the Arctic territories including the Amderma station area where the Hg monitoring was conducted. We underline that the observed increase of the Hg concentration at the station resulted from the regional long-range atmospheric transport of the volcanic cloud (consisting of gases and aerosols) through the Russian Arctic. The arguments are listed in the following paragraphs.

First, in general, during spring period (Mar-Apr-May) the Hg concentrations observed are on average tend to decrease. However, during the Eyjafjallajökull volcanic activity, the corresponding trend was the opposite (**Fig. 4, a**), i.e., towards the higher concentrations in spring. During Apr-May 2010 a large variability in the concentration ($\pm 0.52$ ng m$^{-3}$) was also observed. Apr 2010 showed also the  largest positive trend (+0.97 ng per month).

Second, for the period from May to Jun 2011 the elevated Hg concentrations were also recorded (**Fig. 4, b**). During this

time, the Grímsvötn volcanic activity took place. The positive trend was +1.0 ng per period (being the highest for 13 years of the Hg measurements at the station) with a large variability of $\pm 0.53$ ng m$^{-3}$.



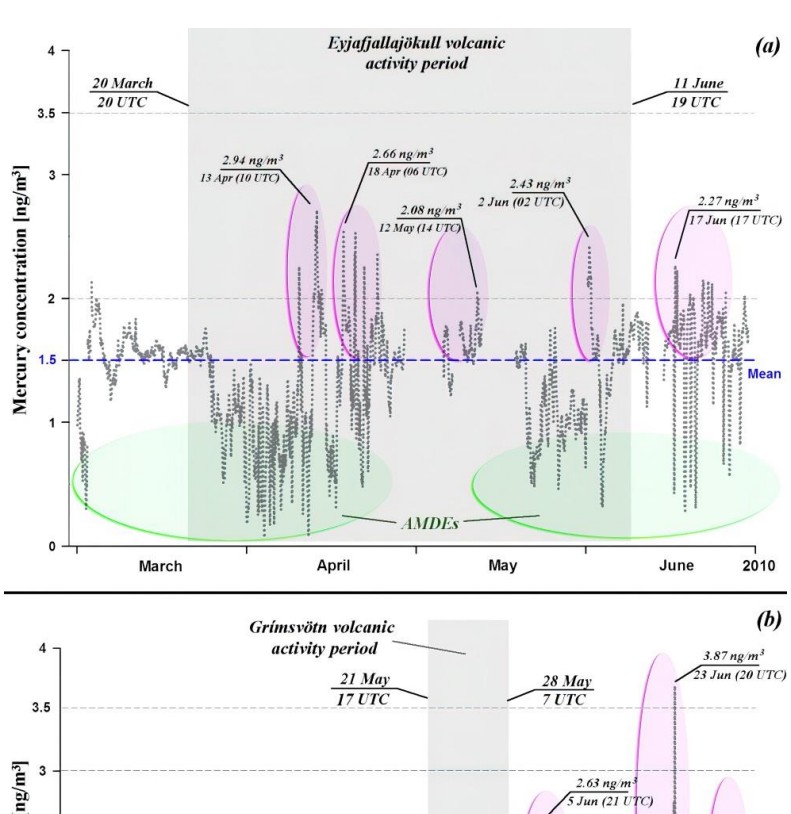

**Figure 4. The observed Hg concentrations at the Amderma station during the eruptions of the Icelandic volcanoes Eyjafjallajökull and Grímsvötn: (a) Mar-May 2010 (b) Apr-Jun 2011. /The green ovals show the atmospheric mercury depletion events (AMDEs). The red ovals indicate the increased Hg concentrations associated with the air masses transport from the areas of the erupting volcanoes/.**

For 2010-2011 (**Fig. 5, a**), the analysis of the time-series on a seasonal behavior revealed existence of the third peak of the Hg elevated concentrations in spring-summer. When the available data do not allow detecting any development trend (trend) due to random and periodic fluctuations in the initial data, the moving average method was applied to better identify the trend (Hyndman, 2009). A moving average is an average value based on subsets of data at specified intervals. Calculating the average value at certain intervals smoothed the data, reducing the impact of random fluctuations or noise. This makes it easier to find common trends within the data. The larger the interval (smoothing window) used to calculate the moving average, the stronger the smoothing occurs, since more data points are involved in each calculated average.

To smooth the yearly series of data, instead of the moving averages, the moving medians were used. This allowed more





clearly demonstrating the nature of data changes without losing the overall trends. The best results were achieved with a smoothing window size of 20 days (**Fig. 5, a**). The ratio of the Hg concentration of the second and third peaks in spring-summer was 0.93 ng m⁻³ (for 2010) and


0.94 ng m⁻³ (for 2011). In the previous (2001–2009) and in the subsequent (2012-2013) years, two peaks of the increased Hg concentration were not observed (**Fig. 5, b**). As a rule, during the spring-summer period only one peak of the Hg elevated concentration was observed. The most likely explanation for such unusual behavior of the atmospheric Hg is the impact of the volcanic eruptions in Iceland.


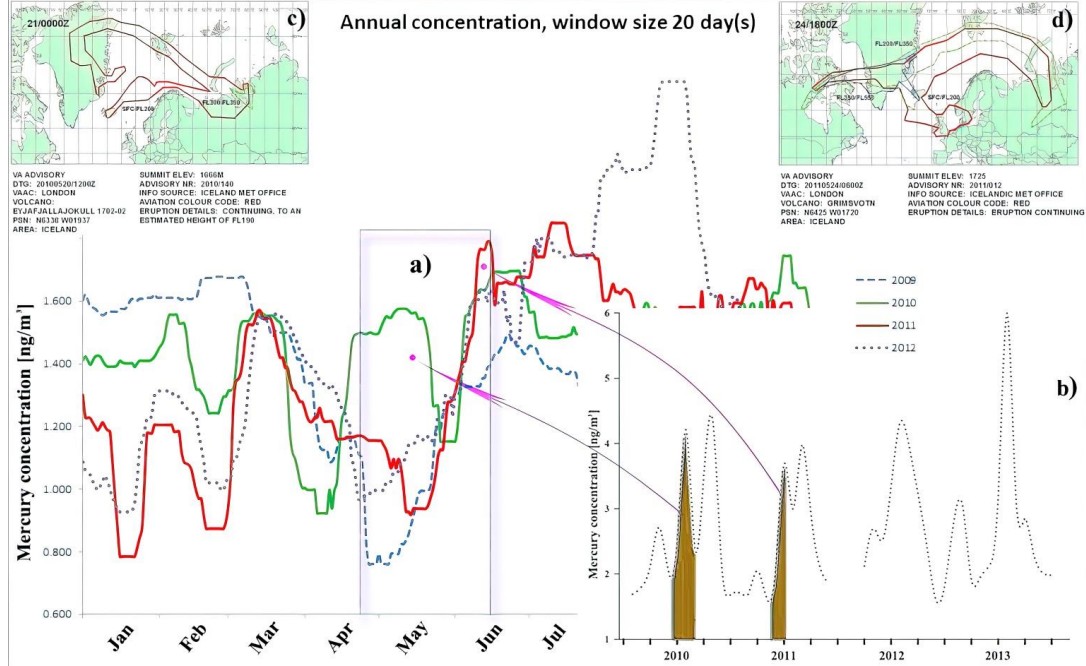

**Figure 5. The behavior dynamics of the mercury concentrations: (a) monthly data 2009-2012; (b) area with high concentration during the volcanic eruption (2 peaks maximum/brown color); and examples of the Icelandic MetOffice data for the volcanos (c) Eyjafjallajökull in 2010, and (d) Grímsvötn in 2011 during active phases of the eruptions.**


The **Fig. 5c, d** (see red lines) shows the enclosed geographical areas where might be high concentrations of volcanic ash and aerosol. The atmospheric transport during the eruption may affect levels of the Hg concentrations in a given geographical region. Consequently, the recorded Hg elevated concentrations during these periods can be connected with the volcanic cloud passage through the area of the Amderma station.

**3.3  Volcanic activity during 2010-2011**

### 3.3.1 Eyjafjallajökull

The elevation of the Eyjafjallajökull volcano is about 1670 meters asl with a crater of about 4 km in diameter. On 20 Mar 2010, the first phase of the eruption took place, and at that time a relatively small amount of the volcanic ash was emitted into the atmosphere (Karlsdóttir at al. 2010). During the second phase of the activity, from 7 to 14 Apr 2010, a large cloud

of the volcanic ash was emitted into the atmosphere up to altitude of 9.5 km (Sigmundsson et al. 2010; Schumann et al. 2011). Later, during 16–19 Apr 2010, according to the Icelandic MetOffice (based on a single C-band Doppler weather radar located in Keflavik) the volcanic emissions of gases and ash reached the altitude of about 8.5 km. Subsequently, the





ash layer was observed at different altitudes around Europe (Ansmann et al. 2010; Flentje et al. 2010; Petäjä et al. 2012; Pappalardo et al. 2013).

The direct shortest distance between the source (the Eyjafjallajökull volcano) and the measurement station (Amderma) is about 3370 km, as seen in **Fig. 1**. There are no large sources of both the anthropogenic and biogenic emissions of Hg into the atmosphere at these high latitudes and along the atmospheric path of the volcanic cloud from Iceland toward the location (Yugor Peninsula, Russia) of the measurement station.

    During the first episode (7-14 Apr 2010) the measured Hg concentration in several cases was more than 2 ng m$^{-3}$. In

particular, it reached on 10$^{th}$ Apr 04 UTC - 2.21 ng m$^{-3}$, on 13$^{th}$ Apr 10 UTC - 2.71 ng m$^{-3}$ and on 14$^{th}$ Apr 2010 06 UTC - 2.07 ng m$^{-3}$. And these concentrations are the highest for the active phase of the Eyjafjallajökull volcano eruption (**Fig. 4, a**). The volcanic cloud was mainly transported in the eastern dominated directions. The cloud moved in the north-eastern direction and passed over the Scandinavian Peninsula, and then, it passed over the Amderma station and surrounding territories. This passage could be the reason for the observed Hg elevated concentrations recorded at the

station. On 16$^{th}$ Apr 2010 (18 UTC) the main atmospheric transport was within the north-west direction and later it turned to the south-west (**Fig. 5, c**). During the second episode (18-24 Apr 2010), the Hg elevated concentrations were also recorded (e.g., on 18 Apr 06 UTC - 2.54 ng m$^{-3}$, on 20 Apr 09 UTC - 2.53 ng m$^{-3}$, on 22 Apr 00 UTC - 2.25 ng m$^{-3}$, and on 24 Apr 2010 09 UTC - 2.36 ng m$^{-3}$). During the third episode (6–12 May 2010), the concentration was also high, i.e., more than 1.82 ng m$^{-3}$. And in particular, it reached on 9 May 20 UTC - 1.82 ng m$^{-3}$ and 12 May 2010 14 UTC - 2.06 ng m$^{-3}$ (as

seen in **Fig. 4, a**).

### 3.3.2 Grímsvötn

    The elevation of the Grímsvötn volcano is about 1725 m asl, and the length of the caldera is about 2 km. The Grímsvötn eruption began on 21 May 2011 (17 UTC). A 21 UTC the volcanic ash released reached the altitude of about 20 km (according to observations of the Icelandic Met Office). The volcanic cloud was transported over Europe and Russia

(Kerminen et al. 2011; Tesche et al. 2012; Moxnes et al. 2014).

    From 23 to 25 May 2011 the cloud passed within the boundary layer over the Yugor Peninsula and the northern territories of the Polar Urals, and then, it continued to move in the north-eastern direction (**Fig. 5, d**). Note, there are no local sources (i.e., industrial facilities) of Hg in the area, and hence, the elevated concentrations could be a result of the volcanic eruption followed by the atmospheric transport towards the Arctic territories including the Amderma station.

The episode with the Hg elevated concentrations took place during the last week of May 2011. At first, such elevated concentrations were observed on 25$^{th}$ May 01 UTC (1.97 ng m$^{-3}$) and on 28$^{th}$ May 2011 00 UTC (1.87 ng m$^{-3}$). Although these values are less than the threshold of 1.87 ng m$^{-3}$, these values can be considered as elevated, because the calculated 13-y e a r average concentration is about 1.51±0.41 ng m$^{-3}$. In particular, similar elevated concentrations were also registered on 29$^{th}$ May 00 UTC (1.78 ng m$^{-3}$) and on 30$^{th}$ May 2011 15 UTC (1.77 ng m$^{-3}$). Starting from 30$^{th}$ May

2011 and later, there were no emissions released to the high altitudes. From 4$^{th}$ Jun 2011, the registration of the Hg elevated concentrations was mainly related to the regional atmospheric transport of Hg (and mainly in a gas phase). At the Amderma station, the elevated concentrations were registered on several occasions (4, 5, 7, 11, 14, 18, and 20 Jun) with the highest value of 3.87 ng m$^{-3}$ observed on 23$^{rd}$ Jun 2011 (see **Fig. 4, b**). Note that Jun–Jul 2011 could be considered as the period with a frequent occurrence of the AMDEs events (Pankratov et al. 2010). However, the registration of the Hg elevated

concentrations at the Amderma station during the polar spring shows that it can be associated with receiving of a large amount of Hg from a strong source, which in our case could be the Icelandic volcano in the active phase of eruption.

### 3.4 Trajectory analysis connecting the volcanic activity and observation

    Our analysis of the calculated backward trajectories showed that, indeed, the eruption of the Eyjafjallajökull volcano could



influence the increase in the Hg concentrations in the surface layer of the atmosphere at the Amderma station. The
trajectories were calculated for the three altitudes (500, 1500 and 3000 m asl) and backward in time up to 5 days. Note that
these trajectories were calculated for the time period when the volcanos were in the active phase of the eruption.

The trajectories were calculated for selected dates (13 and 18 Apr, and 1 Jun 2010) when the Hg concentration was
considered as the elevated, about 2 ng m$^{-3}$ (see **Fig. 6, a**). From Apr to May 2010, according to our data, we can assume
that the observed Hg elevated concentrations at the station Amderma are due to the long-range atmospheric transport of the
volcanic cloud of gaseous and particulate fractions of components formed during the active phase of the Eyjafjallajökull
eruption (see **Fig. 6, a**).

From May to Jun 2011, the increase in the Hg concentration also resulted from the long-range atmospheric transport, but in
this case from the eruption of the Grímsvötn volcano (see **Fig. 6, b**). For selected dates (25 May, 5 and 23 Jun 2011) the
calculated trajectories showed arrival of the air masses from the higher altitudes (between 1–6 km), and it is typical for the
global atmospheric transport. For 25$^{th}$ May 2011, the atmospheric transport showed movement of the air masses in the
middle troposphere at the altitudes between 1–5 km. Note, that in Jun 2011 the air flow differed in the altitude profiles
characteristic for the hemispheric scale atmospheric transport in the middle layers of the troposphere (see **Fig. 6, b**).

It can be confirmed that the Hg elevated concentrations co-inside with the atmospheric transport from the north-west and
linked with the active volcano Grímsvötn. For 23$^{rd}$ Jun, the atmospheric flow, moving above the Novaya Zemlya
Archipelago (Russia), arrived at the Yugor Peninsula from the north- east direction (similar to the Met Office UK
calculations).

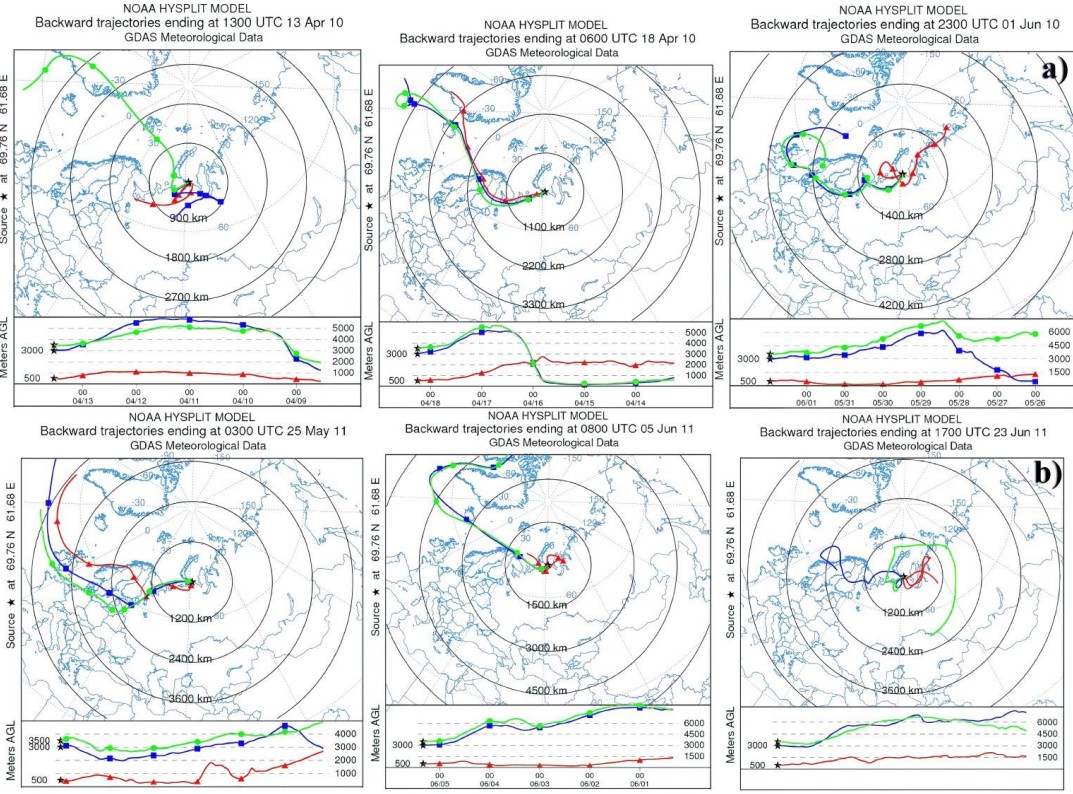

**Figure 6. The atmospheric backward trajectories (calculated by the NOAA HYSPLIT model) arriving at the Amderma station
on (a) 13, 18 Apr and 1 Jun 2010 and (b) 25 May and 5, 23 Jun 2011 and corresponding to the eruptions of the (a)
Eyjafjallajökull and (b) Grímsvötn volcanoes.**



Finally, it can be assumed that the volcanoes are the large natural sources of Hg as during the active phase of the eruption, and possibly also during the degassing, when there are no intense emissions. In our case, considering that the Eyjafjallajökull and Grímsvötn volcanoes are the main sources of Hg in the Northern Hemisphere, the intake of Hg in the Arctic ecosystems can be calculated using the Hg concentration measurement data following the periods of the active volcanoes' eruptions. Moreover, Sönke et al. (2017) showed that a lot of mercury is transported to the Arctic via various rivers in water soluble form.

## 4 Conclusions

In this study, we performed a detailed analysis of the long-term measurement data of atmospheric mercury concentrations in the surface layer of the atmosphere. We linked the elevated concentrations with the periods of the eruptions of the two volcanoes Eyjafjallajökull and Grímsvötn (Iceland). Our analysis of the long-term (2001-2013) time-series of the Hg concentrations underlined the time periods with the lower and higher concentrations at the Amderma station in the Russian Arctic. Our key findings are summarized as follows:

- For 2001–2009 period, the annual average concentration of Hg decreased from $1.67\pm0.31$ (2001) to $1.32\pm0.31$ ng m$^{-3}$ (2009) with the negative trend identified as -0.66 ng per period. However, the last three-year period (2010–2012) in the time-series showed the highest positive trend (+0.97 ng per period).

- During Apr-May 2010 and May-Aug 2011 (the periods of the Icelandic Eyjafjallajökull and Grímsvötn eruptions) the Hg concentrations were significantly higher, or elevated, and in particular: $1.81\div2.71$ ng m$^{-3}$ and $1.81\div3.69$ ng m$^{-3}$, respectively. These values are higher compared with the average value of $1.51\pm0.41$ ng m$^{-3}$ for the Russian Arctic in the Northern Hemisphere.

- Application of the moving average method to the time-series showed a good correlation between the dates of the volcanic eruptions in Iceland and the registration of the elevated Hg concentrations at the Amderma station.

- During the polar spring the registration of the low concentrations of Hg in the surface layer of the atmosphere was also observed. It is known as the atmospheric mercury depletion events, AMDEs. Therefore, the presence of the elevated Hg concentrations is not typical for spring (observed in 2010 and 2011), and the increased Hg concentrations in this Arctic region are determined by the atmospheric transport at the regional and hemispheric scales.

- The calculated atmospheric backward trajectories showed that the source of the elevated Hg concentrations in this region of the Arctic are connected with the active phases of the volcanic eruptions in Iceland. At the time of the eruptions of Eyjafjallajökull and Grímsvötn volcanoes, many meteorological models also showed the atmospheric transport of the volcanic cloud over territories of the Russian Arctic including the Yugor Peninsula. Therefore, the gaseous mercury deposited on aerosol particles could influence the increase in concentrations measured at specific locations (e.g., Amderma station). Hence, the Hg inflow into the Arctic during the two Icelandic volcanic eruptions occurred basically due to Hg presented in the gas phase as well as due to the fractions of Hg deposited on the aerosol particles.

Overall, our analysis indicates that the registration of the elevated Hg concentrations at the Amderma station in 2010 and 2011 is an anomaly in terms of the dynamics of the global atmospheric mercury pollution, and it occurred due to the volcanic eruptions in Iceland and peculiarities of the atmospheric transport in this part of the Arctic.

## 5  Acknowledgements

The authors gratefully acknowledge colleagues - Strelnikov I., Kozulin S., and Balandin V. - for invaluable assistance, continuous monitoring and maintaining the Tekran mercury analyzer during 2001–2010, as well as Mikushin A. and

...



Guskov V. (technical staff of the hydrometeorological service) during 2011-2013 at the Arctic station Amderma (Russia). The authors gratefully acknowledge the NOAA Air Resources Laboratory (ARL) for the provision of the HYSPLIT transport and dispersion model and/or READY website (http://www.arl.noaa.gov/ready.html) used in this publication.
Financial support for the monitoring program was provided by Environment Canada, Arctic Monitoring and Assessment Programme (AMAP) Secretariat and Russian Federal Service for Hydrometeorology and Environmental Monitoring (Roshydromet). For this study, the financial support was also partially provided by the Kola Science Center of the Russian Academy of Sciences and the Pan-Eurasian EXperiment (PEEX; https://www.atm.helsinki.fi/peex) programme and the European Union's Horizon 2020 research and innovation programme under grant agreement No 689443 via project iCUPE (Integrative and Comprehensive Understanding on Polar Environments; https://www.atm.helsinki.fi/icupe).

**Author contribution.** FP - designed and carried out the mercury measurements with technical support, analysis and formalization data, writing draft of the manuscript. AM - co-writing draft of the manuscript, comments and remarks with contributions from all co-authors. OK - data formalization, calculation algorithm, data post-processing. TP – comments and remarks with contributions from all co- authors. VP - comments and remarks with contributions from all co-authors. VM - comments and remarks with contributions from all co-authors.

**Competing interests.** The authors declare that they have no conflict of interest. There are no competing interests. Prof. Tuukka Petäjä is an Editor of Atmos Chem. Phys.

**Institutional Review Board Statement:** Not applicable.

**Informed Consent Statement:** Not applicable.

**Data Availability Statement:** Data supporting reported results can be found as "Dataset on Long-term monitoring of gaseous elementary mercury in background air at the polar station Amderma, Russian Arctic" at http://www.atm.helsinki.fi/icupe/images/Datasets/DS_Hg-Amderma_20200125.zip (last access on 27 January 2025),and readme file at https://www.atm.helsinki.fi/icupe/images/Datasets/Readme_iCUPE-Collabor_DatasetReleased_Hg.pdf (last access on 27 January 2025), and at the ZENODO portal - Fidel Pankratov (2020). Long-term monitoring of gaseous elementary mercury in background air at the polar station Amderma, Russian Arctic (Version 1) [Dataset]. Zenodo. https://doi.org/10.5281/zenodo.4060211 (last access on 27 January 2025).

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
