# Peer review of "Measurement report: Atmospheric mercury measurements at the Russian Arctic station Amderma and connection with eruptions of Icelandic volcanoes"

_EGUsphere, 2025_

## Author Comment (AC2)

**Dear Anonymous Reviewer #2,**

Thank you for your constructive comment, suggestions, remarks. Please, find below the answers on these. Respectfully, on behalf of co-authors, Fidel Pankratov.

The manuscript presents a long-term analysis (2001–2013) of atmospheric mercury concentrations at the Amderma Arctic station, aiming to link observed enhancements in 2010 and 2011 to volcanic eruptions in Iceland. The dataset is valuable, and the integration of continuous monitoring with backward trajectory analysis provides a potentially novel insight into long-range transport of volcanogenic mercury. However, several important issues need to be addressed before the study's conclusions can be fully supported. These include overinterpretation of seasonal trends, lack of statistical testing for trends, and insufficient discussion of 2012 peak.

Specific Comments

1. The introduction is too long and some sentences are off topic. For example, the paragraph starting as "Some studies show that continuously degassing…." (lines 70-81) can be removed because the role of volcanic emissions is already introduced earlier. This paragraph shifts focus to other volcanic gases, which are outside the main scope of this study. Also, the sentences with lines 94~96 show the indirect comparison to Bi is not used in this result or discussion. It adds unnecessary complexity. I suggest that the authors read the manuscript thoroughly and effectively reduce the introduction part.

   A.: This part of the Introduction section was fully re-written and shorten to take into account/ consider the reviewers comments: sentences off-the-topic were excluded/ removed, information about other volcanic gases was also removed, and by removing the unnecessary complexity was minimized, leading to reduction to the Introduction-section.

   70    … (Fendley et al. 2024). Accordingly, the question arises whether passive degassing can trigger some of the processes that can lead to an eruption (Girona et al. 2015). The average contribution of $SO_2$ into the atmosphere by volcanoes, estimated largely by extrapolation from direct measurements of volcanic SO2, is about 18 t $yr^{-1}$. The estimated 6 t $yr^{-1}$ (36%) is from non-erupting degassing volcanoes (Stoiber et al. 1987). $SO_2$ contributes 64% to the global volcanic sulfur flux of 10 t $yr^{-1}$ (Andres and Kasgnoc, 1998). However, we can use the directly measured $SO_2$ fluxes and known molar ratios (e.g., $H_2S/SO_2$) for a semi-quantitative estimate of other gas components emitted (e.g., $H_2S$). The total annual emission of HCl is 1÷170 t $yr^{-1}$ (Halmer et al. 2002). After correcting for "unmeasured" $SO_2$ emissions, the estimated total global flux of Hg into the atmosphere is about 112 t $yr^{-1}$. …

2. Methodology: There is very detailed description of the Tekran analyzer, most of which is best referred to as supplementary material.

   We would like to keep detailed description in the Methodology-section, because it is shortened compared with official website (link: (https://www.tekran.com/products/ambient-air/tekran-model-2537-cvafs-automated-mercury-analyzer/) description of the Tekran gas-analyser, and one of the reviewers suggested to provide even more detailed description compared with the original one text.

   123    … (2) Tekran Model 1100 Zero Air Generator; (3) Tekran Model 1120 Standard Additional Controller; and (4) Automated Ambient Air Analyzer "Tekran 2537A" (https://www.tekran.com/products/ambient-air/tekran-model-2537-cvafs-automated-mercury-analyzer). …

3. The manuscript reports a decreasing trend in Hg concentrations during the 2001-2010 period and an increasing trend during the 2010-2012, but this conclusion appears to be drawn based on visual inspection of the time series and the average values rather than a formal statistical analysis. I recommend that the authors support this claim with appropriate statistical methods.

   The following text was added in section:
   "In our study we utilized the SigmaPlot statistical package. At first, a test for normality of the collected data (time-series of mercury concentrations) was performed. It is assumed for all parametric tests and regression related procedures that makes assumptions about the population parameters. It showed (see below) that during both periods 2001-2009 and 2010-2012 the mercury concentration shows distribution close to the normal (see Figure A1 in Annex), and therefore, calculation of standard statistics (monthly and seasonally: average and median in Figure 6 & standard deviation and variance in Figure A2 in Annex) is applicable.

The applied method is the method of regression analysis, and in particular, the simple linear regression. We focused only on analysis of concentration of mercury (given as numeric values measured on a continuous scale using number) and its dependence on time. To calculate trends for two selected periods in years of 2001-2009 and of 2010-2012, the fitting a curve as straight line through the observation data (mercury concentrations) was done for both periods. It allowed to find the slope and the intercept of the constructed line y=p0+p1x. Such line most closely describes the relationship of observations, where y as concentration (the dependent variable) and x as a time (the independent variable). These are summarized in Table A1 in Annex."

4. Figure 2. What is the time resolution of the measurement values? Daily concentrations? Please describe the time resolution in the caption of this figure. Also, this figure omits some of the extreme high values that are mentioned in the text. While adjusting the y-axis scale improves visibility of the general trend, I suggest indicating in the figure caption that these outliers were clipped. Alternatively, consider adding an inset plots showing the full range of values to preserve data transparency. Sometimes, outliers are very important.

The mercury concentrations in ambient air were measured at 30-minute intervals across two channels. After verifying the data quality, an average of two consecutive measurements (1-hour averages) was calculated, along with daily average mercury concentrations. Sampling, preparation, and analytical methods for atmospheric mercury monitoring were implemented based on EPA Method 1669.

The caption to Figure 2 to take into account the comments above was re-written as follows:

"Figure 2. The long-term time-series of hourly measurements of the atmospheric mercury concentration … For improving visibility of generated trend, the outliers (e.g., values with concentration above as seen 4.5 ng m$^{-3}$) in time-series were clipped"

5. Figure 3. The authors present seasonal trends of Hg concentration, but the specific month ranges used to define each season are not explicitly stated. Please, clarify it.

The seasons are difined as: Spring season: March-April-May; Summer season: June-July-August; Autumn season: September-October-November; and Winter season: December-January-February.

6. While the authors emphasize an increase in mercury concentrations during 2010–2012, **Figure 3** does not clearly support this claim. In fact, seasonal concentrations in 2010 and 2011 appear comparable to or even lower than some years in the earlier 2001–2008 period. The observed rise may largely reflect a dip in 2009 rather than a significant upward shift. I recommend conducting a statistical comparison between the 2001–2009 and 2010–2012 periods (e.g., using seasonal mean tests or trend analysis) to robustly support the claim of anomalous increases related to volcanic activity.

For months of spring of 2007-2013 (as years before – during – after) the eruption are divided into 3 ranges of concentration: less than 1, within interval 1.5-1.8, and more than 1.8 ng m$^{-3}$ and summarized in **Table 2**. Moreover, Figure 6 underlines average and median month-to-month and seasonal variabilities of mercury concentration for 2001-2012 (and **Figure A2** in Annex – for standard deviation and variance). **Table A1** in Annex summarizes trend lines equations by time periods, by months within periods, and by seasons within periods during 2001-2012, 2001-2009, and 2010-2012 mercury measurements at the Amderma station.

7. The manuscript attributes elevated mercury concentrations during 2010–2012 to volcanic eruptions in 2010 and 2011. However, **Figure 4** shows that the highest Hg concentrations were observed well after the main eruption period, particularly in June 2011. Moreover, **Figure 3** indicates that 2012 had the highest seasonal mean concentrations, yet the authors provide no specific explanation for this anomaly. The authors mentioned that the registration of the Hg elevated concentrations was mainly related to the regional atmospheric transport of Hg from 4[th] Jun 2011 (line 340), but it is just blanket claim, and there is no specific explanation for the particularly high concentrations in 2012 (summer and autumn).

The main period of eruption of Eyjafjallajokull volcano took place during 20 March – 11 June 2010 (as seen in **Figure 4a**) with maximum concentrations measured on 18 April – 12 May – 2 June 2010 and also

higher concentration until 17 June 2010. The main period of eruption of Grimsvotn volcano took place during 21-28 May 2011 (as seen in Figure 4b) with maximum concentration measured on 5 June – 23 June – 1 July 2011. In 2012, the Hg concentration showed episodically vales higher than 2 ng m$^{-3}$ as seen in Figure 3. It should be note that the volcanoes exhibited prolonged residual degassing, remaining a source of transport of fractions in both gaseous and solid phases (persisting until, at least, 2012). Secondary re-emission of Hg deposited by the volcanic plume is plausible, but there are no direct studies conducted to confirm this process and elevated Hg concentration at the monitoring site.

8. Looking at **Figure 5**, it seems clear that the Eyjafjallajokull eruption caused an increase in Hg concentrations in April-May 2010. However, when comparing the concentration trends in 2011 with 2012, it seems difficult to see the impact of Grimsvotn on Hg concentration in May and June in 2011.

   For clarity, the horizontal axis was improved in **Figure 5a** with making intervals accordingly. This Figure shows that the volcanic plume reached the monitoring site following the eruption's active phase. Analysis presented in **Table 2** shows that the third peak (seen as green circle, in **Figure 5a**) in late spring 2010 (in particular, in May) was identified and such peak (seen as red circle, in **Figure 5a**) was also identified in early summer 2011 (in particular, in June) that underlines impact of the Eyjafjallajokull and Grimsvotn volcanoes eruptions, respectively. These third peaks in May 2010 and June 2011 are not driven by surface re-emissions from the underlying surface occurred annually, but these peaks are most probably attributed to the Icelandic volcanic eruptions.

9. Section 3.4. The backward trajectory analysis in Section 3.4 focuses only on days with elevated Hg concentrations. However, to robustly support the link between volcanic activity and increased Hg levels, it is essential to assess whether similar air mass trajectories occurred on days without elevated concentrations. If the synoptic-scale atmospheric circulation during the study period tended to bring air from similar source regions regardless of Hg levels, the observed coincidence might be incidental rather than causal.

   Two additional subplots – **Fig. 7c,d** as newly numbered and compared to older version as **Figure 6** - were added to this figure. These 2 additional subplots with calculated trajectories showing examples of specific dates/times for non-elevated Hg concentrations at the Amderma station when eruptions did not take place

[Figure]

The dominated synoptic-scale circulation did not tend to bring air masses as there were no similar regions (in Northern Hemisphere) with locked sources of mercury, such as volcanic eruptions in 2010 and 2011. We consider the observations of Icelandic MetOffice (based on single C-band Doppler weather radar located in Keflavik) and the routes constructed by them for the transport of volcanic clouds during volcanic eruptions to be confirmation of local transport. **Fig.5c,d**. In this case, there is a good correlation between the atmospheric paths of the volcanic cloud from Iceland toward the location (Yugor Peninsula, Russia) of the measurement station.